# Additive Manufacturing of Plastics Used for Protection against COVID19—The Influence of Chemical Disinfection by Alcohol on the Properties of ABS and PETG Polymers

**DOI:** 10.3390/ma14174823

**Published:** 2021-08-25

**Authors:** Krzysztof Grzelak, Julia Łaszcz, Jakub Polkowski, Piotr Mastalski, Janusz Kluczyński, Jakub Łuszczek, Janusz Torzewski, Ireneusz Szachogłuchowicz, Rafał Szymaniuk

**Affiliations:** Institute of Robots & Machine Design, Faculty of Mechanical Engineering, Military University of Technology, 2 Gen. S. Kaliskiego St., 00-908 Warsaw 49, Poland; krzysztof.grzelak@wat.edu.pl (K.G.); julia.laszcz@student.wat.edu.pl (J.Ł.); jakub.polkowski@student.wat.edu.pl (J.P.); piotr.mastalski@student.wat.edu.pl (P.M.); jakub.luszczek@wat.edu.pl (J.Ł.); janusz.torzewski@wat.edu.pl (J.T.); ireneusz.szachogluchowicz@wat.edu.pl (I.S.); rafal.szymaniuk@student.wat.edu.pl (R.S.)

**Keywords:** additive manufacturing, structural analysis, mechanical properties, polymers, COVID19

## Abstract

In this paper, the influence of disinfection on structural and mechanical properties of additive manufactured (AM) parts was analyzed. All AM parts used for a fight against COVID19 were disinfected using available methods—including usage of alcohols, high temperature, ozonation, etc.—which influence on AM parts properties has not been sufficiently analyzed. During this research, three types of materials dedicated for were tested in four different disinfection times and two disinfection liquid concentrations. It has been registered that disinfection liquid penetrated void into material’s volume, which caused an almost 20% decrease in tensile properties in parts manufactured using a glycol-modified version of polyethylene terephthalate (PETG).

## 1. Introduction

COVID19 pandemic caused a massive gap in deliveries of first-aid tools for human life and health protection against the virus. At the beginning of lockdown in 2020 all companies, freelancers, and hobbyists connected with additive manufacturing (AM) of polymers started to produce face shields, diving masks high-efficiency particulate air (HEPA) connectors, anti-dust masks HEPA connectors, and other different tools.

Material extrusion additive manufacturing (ME-AM) plays a significant role in many industries because of its advantages, such as easy digitization based on three-dimensional CAD data, fast, and low-cost efficient creation of customized, on-demand, and prototype products, and reflecting accurately very complex geometries. One of the most common and popular methods is fused filament fabrication (FFF), where thermoplastic material (filament) is heated in a range of 175–265 °C (depends on material type). The plasticized filament is being deposited layer by layer in the building plate in such a way to allow reaching the desired shape of the same part. There is a variety of filaments used in this technology. Poly(lactic acid) (PLA) and acrylonitrile butadiene styrene (ABS) are the most characteristic materials used in that type of ME-AM. PLA is non-toxic, biodegradable, biocompatible therefore, it is used in the field of medicine [1,2,3,4,5]. However, objects made of PLA are characterized by low mechanic properties [6], especially at a temperature near plasticizing point (about 60 °C) [7] and are influenced by low fatigue resistance, too [4]. That is why this material has not been used during AM of parts dedicated for protection against COVID19 [2].

Many researchers analyzed the influence of FFF technology parameters (layer thickness, infill density, and pattern) on the final product properties and used the material itself. One material commonly used is polyethylene terephthalate glycol (PETG), which is more flexibly and resistant to temperature than PLA, and has high durability, low shrinkage, and is hydrophobic [8]. These specific properties of polyethylene terephthalate glycol led to it being chosen as the sufficient material for 3D printing of masks during the beginning of the COVID-19 pandemic [3,9,10]. Srinivasan et al. [11] studied the mechanical properties of PETG. Their research noted that the tensile strength of the printed models is directly proportional to infill density and inversely proportional to layer thickness. Additionally, it is characterized by reverse relations in surface roughness. Durgashyam et al. [12] investigated PETG materials not only tensile but also flexural strength while considering the specifications mentioned above. The main conclusion was that, among tested parameters, layer thickness had the highest contribution in affecting mechanical properties of produced objects.

On the one hand, Hanon et al. [13], after evaluating the outcome of their analyses deduced that PETG used in FFF technology demonstrates anisotropic properties, but on the other hand, Mercado-Colmenero et al. [14], after complex research proved that in numerical simulations FFF manufactured PETG could be treated as an isotropic material.

Since the world is currently struggling with COVID19, AM has been exploited to the boundaries of its possibilities. In medicine, it filled significant gaps where traditional subtractive manufacturing is lacking, injection molding is not ready for mass production, or is economically unjustified. AM is a technology in which the product is constructed by adding material in cross-sectioned layers, this way of processing influences the fatigue performance of the element.

The main reason for using PETG for the health protection during the beginning of the COVID19 pandemic is two-fold. The first is related to the material properties—it is more resistant to temperature than the PLA—it can withstand temperatures up to 75 °C and its properties are not affected by the UV radiation. Also, it is characterized by better chemical resistance than ABS and PLA and is less fragile than the PLA parts. Such characteristics allow bettering disinfection using high temperature or alcohols which made it better customized to medical solutions than the PLA and other commonly used materials in ME-AM. On the other hand, the usage of the PETG at the beginning of the COVID19 pandemic was related to low filament diameter tolerance increase after winding increase (prusament.com) to allow the increase of manufactured material dedicated to FFF technology.

Additionally, there are available in the market, materials dedicated for the ME-AM, and usage in medical applications. The most popular are ABS-based filaments. Such materials are mostly certified by basing on USP VI and ISO 10993-1 standards. Its usage during the pandemic was marginalized due to its high cost and low availability.

Nowadays, many Universities and companies joined forces to produce better protection for the medical staff as they are the most endangered. AM has been used chiefly to create protection helmets as it is a fast way to build many specific parts [15] or connectors for HEPA filters. This equipment needs to be disinfected regularly to avoid introducing pathogens, which are the cause of the illness. Low-level disinfection (LLD) is based on exposing the surface to a liquid for at least 1 min. The substance used is a fluid that contains 70–90% of alcohol [16]. The aftermath of its effect should be the annihilation of viruses, vegetative bacteria, and some fungi. Going further, high-level disinfection (HLD) results in destroying all microorganisms except some bacterial spores with heat treatment performed by subjecting the material to 65 to 75 °C for half an hour [16].

Based on Huang et al. work [17] the main categories of chemical cleaning agents can be listed as follows: acids, alkalis, disinfectants (including alcohols), surfactants, metal chelating agents, and enzymes. The authors of the mentioned work [17] discussed some most important cases of disinfection mechanisms:2.2-Dibromo-3-nitrilopropionamide inhibits respiration and inactivates proteins containing nucleophilic partial amino acids such as methionine and cysteine.Isothiazolones quickly inhibit the physiological functions of microorganisms, including growth, respiration, energy production (such as adenosine triphosphate synthesis), and destroy thiol-containing proteins.Glutaraldehyde reacts with biomolecules like protein, RNA, and DNA, which contain amino, amide, and carboxyl groups.Tributyl tetradecyl phosphonium chloride inactivates bacteria by destroying and decomposing the negatively charged membrane of bacteria.Dichloroisocyanurate releases hypochlorous acid and isocyanuric acid in water to inactivate bacteria.Copper and silver ions interfere with enzymes involved in cell respiration and bind to DNA at specific sites.Ethyl lauroyl arginate inactivates microorganisms by changing their cell membrane structure and interfering with their membrane potential.Chlorhexidine gluconate acts as a biguanide and cation-active compound with significant antibacterial activity and inhibits microorganism adherence and prevents biofilm formation.

To render more stringent the case of polymers chemical disinfection, Roman et al. [18], analyzed different types of disinfection that could be used for AM parts. The authors revealed five different methods which could be used for such parts:ultraviolet (UV) sterilization using a germicidal fluorescent bulb,autoclave sterilization,submersion in a glutaraldehyde solution,hydrogen peroxide sterilization,alcohol disinfection.

Based on mentioned research [18], hydrogen peroxide sterilization is the best alternative to avoid the deformation of PLA and PETG AM objects. It is also preferable to steam sterilization for PLA and PETG, as it causes only submillimeter morphological distortions, instead of dynamically damaging the materials in consequence of high temperature (121 °C for 5 min) [19].

Electron beam and gamma radiation are also a practice in the mentioned above field and are safely used for PLA and PETG. However, ethylene oxide cannot be implemented for PLA and PETG, as it inflicts the polymeric structures, causing weight loss and creating a risk of inducing toxic deposits on the surface of the element [19]. On the other hand, UV and gamma radiation does not impact the alignment of the fibers, therefore are suitable for PLA sterilization [20]. ABS disinfection is not an often mentioned topic in studies but hydrogen peroxide works very well with it, not altering the morphology of the filament [21]. The abovementioned methods were not available in smaller facilities that struggled with the effects of the COVID19 pandemic. Despite many available methods, the most available disinfection medium for most medical aid centers and social welfare facilities was alcohol. Additionally, the usage of disinfection liquid allows better penetration of the whole volume of each part. It is especially important in AM parts which are often characterized by geometrical complexity.

As shown, in most of the cited studies conducted over the past two years, the mechanical properties of PETG are still being investigated. Still, none of them has included LLD or HLD influence on mechanical properties of FFF manufactured PETG models such as face shields widely used by rescue teams and hospital staff during the pandemic. Usage of the PETG in FFF technology seems to be a good alternative in some solutions where much more expensive AM technologies are unnecessary. This statement results from our previous research, connected with other material also dedicated for medical solutions–316L steel obtained during selective laser melting (SLM) process [17,18,19,20,21].

## 2. Materials and Methods

The FFF method produced the test specimens using AM technology. The main reason for selecting the FFF technology was using that kind of device during the 3D print of many different parts dedicated to increasing protection against COVID19. The principle of the FFF process is very similar to fused deposition modeling (FDM) technology, where the plastic wire (named as 3D printing filament) is being pushed by extruder mechanism to the heated nozzle. After that, the material is being plasticized and put into the substrate plate. A fully automated movement algorithm of the printing nozzle and substrate plate allows for different (often very complex) shapes creation. Regarding the additive character of the process, the parts are built layer-by-layer until the final geometry creation.

For the research PETG filament (Rosa 3D, Hipolitów, Poland) was used. That kind of material is prevalent for manufacturing parts dedicated to protection against COVID19. It was also commonplace for using different colors of the material during many supporting actions. Hence, two types of PETG were tested: with color pigment and without any additions. All parts were manufactured using the Prusa Original MK3s device (Prusa Research, Prague, Czech Republic), the manufacturing processes were prepared using PrusaSlicer Software (version 2.1.0) using the default parameters for the Prusa device for PETG materials:Hotend temperature (for both materials): 240 °C,Heatbed temperature: 90 °C,Layer thickness: 0.2 mm,Infill: 100%,Part cooling intensity: 40%,Printing speed: 60 mm/s,Nozzle diameter: 0.4 mm,Number of the contour lines: 5

Additionally, an ABS Medical (SmartMaterials 3D, Jaén, Spain) filament was used for results compared with the PETG results. That kind of material is certified by basing on USP VI and ISO 10993-1 standard, which assures its biocompatibility. The properties of used materials are shown in Table 1.

A manufacturing process with the usage of the ABS Medical was prepared in the same software and manufactured on the same device as PETG parts. Process parameters–default for the Prusa device for the ABS material were as follows:Hotend temperature: 255 °C,Heatbed temperature: 110 °C,Layer thickness: 0.2 mm,Infill: 100%,Part cooling intensity: 25%,Printing speed: 60 mm/s,Nozzle diameter: 0.4 mm,Number of the contour lines: 5

Based on “ASTM D638: Standard Test Method for Tensile Properties of Plastics” dog-bone-shaped parts were made to determine the mechanical properties. For each test, five same samples were prepared to keep mechanical properties results reliable.

For structural analysis, Keyence VHX 7000 optical microscope (Keyence International, Mechelen, Belgium) was used. All parts were observed using an additional scattered light from the bottom side of the elements.

Axial tensile strength tests were carried out using the Instron 8802 (Instron, Norwood, MA, USA) hydraulic pulsator using an extensometer with a measuring base of 50 mm. During tensile testing, a digital image correlation (DIC) (non-contact, optical method) was used to measure three-dimensional (3D) deformations from Dantec Dynamics (Dantec, Ulm, Germany).

All manufactured parts were divided into four different groups for the HLD process, each group was characterized by different disinfection times. There were four groups:0.5 h disinfection,12 h disinfection,24 h disinfection,48 h disinfection.

To reach desirable results, a Gigasept Instru AF (Schülke & Mayr GmbH, Norderstedt, Germany) disinfection liquid was used. Parts were put into two disinfection can–with 4% liquid solution with water and 100% liquid. After the process, all elements were drained into the laboratory dryer at a temperature of 45 °C for one hour. Such temperature was selected to be at the safe level below the glass temperature of the PETG (which is 70 °C). All samples were held at the same temperature. Additionally, to allow better analysis, all disinfected parts were compared to non-disinfected reference samples. All combinations with sample type descriptions are shown in Table 2.

## 3. Microscopical Investigation—Results and Discussion

To properly describe the microstructural phenomena during HLD all registered images for each sample were compared in Table 3 and Table 4. There is a visible effect of disinfection with two different substance concentrations: 4% and 100%. Also, a comparison with nondisinfected samples (0%) was made. Investigating samples under a light microscope has shown no significant impact (like melting or rinsing of the material) on samples’ structure.

In the case of PETG samples held in 4% solution (Table 4), there is an outer contour and the infill visible, between these two parts of geometry there are visible some brighter areas—especially in samples after 12-, 24-, and 48-h during disinfection. This can be seen as intense flashes in the picture. Materials used in this study like most materials being used in FDM printing are known for their porous structure. This leads to the conclusion that during the disinfection process the liquid is penetrating the structure of the sample and does not evaporate during the drying process, as a result, it may stay inside longer. In the case of the ABS material, the effect of penetrated water inside the object is much less noticeable. It could be due to better filling of materials or, basically, its hygroscopicity.

At 100% concentration (Table 4) brighter areas in PETG are more visible than at 4% concentration. Such phenomenon is connected to the lower dilution of used liquid. Perhaps it may be caused by the high viscosity of not diluted concentrate. This leads to the conclusion that during the disinfection process the liquid is penetrating the structure of the sample and in cases of lower water content in used formulas, the liquid does not evaporate during the drying process, as a result, it may stay inside longer. Also, in the case of 100% solution, samples made of the ABS material has no visible liquid inside their volume.

Accumulating high concentrations of disinfection liquids in structures of FDM/FFF based face shields may ensure protection from some microorganisms but on the other hand, it may also lead to aggregation of dirt or in worse cases, raising of bacterial biofilm, by creating a good habitat for pathogens, or else come into reaction with the component, which might be harmful to the user, this may indicate using different materials in the production of medical helmets or additional covers (for example resin-based).

## 4. Tensile Testing and DIC Analysis—Results and Discussion

To better understand the influence of HLD on AM parts, tensile tests were made. All samples have been additionally analyzed by using DIC. A representative course of samples from each group is shown to allow a more straightforward interpretation of the test results. Strain–stress curves of PETG without pigment in three conditions: 4% liquid solution with water, 100% liquid, and without any disinfection (P0) are shown in Figure 1.

All types of HLD in 4% negatively affect tensile properties, especially in the case of samples that were kept in disinfection solution for 48 h–where tensile strength decreased from 53.98 MPa to 45.11 MPa, which is about 20%. More significant differences were registered for samples subjected to HLD in 100% concentrate. The most significant decrease was reported for samples kept in the disinfection for 12 h and 48 h, and the value was about 25%. Results of the obtained DIC analysis are shown in Table 5. 

The material without any HLD is characterized by increased strain areas and more than one spot of fracture initiation. In most samples subjected to different types of HLD, there is only one spot of fracture initiation. Additionally, in some HLD cases, there is no visible strain increase along X-axis. That phenomenon could relate to a negative influence of the disinfection liquid, which came into the material volume between the infill and outline perimeter shell.

The same type of analysis was conducted for colored PETG material. Stress–strain curves for all tested samples, including reference material without any disinfection, are shown in Figure 2.

In the case of colored PETG, there is also visible a slight decrease in tensile strength of samples subjected to HLD, but it is lower than 5% compared to non-disinfected samples. A base for the pigment addition could increase the material’s chemical resistance and lower the influence of a disinfection liquid. That hypothesis could be proven by the DIC analysis shown in Table 6.

Regarding obtained DIC results, there is a visible a different behavior of the material after HLD in 4% water solution and 100% concentrate. After disinfection in 4% water solution, there was an equal strain distribution, which cannot be concluded in samples that were sunk in 100% concentrate. Also, those samples have a visible increase in strain across the X-axis (especially in the breaking point) in comparison to samples held in a 4% water solution. It could be related to the significant concentration of the disinfection liquid, which did not affect that kind of phenomenon in 4% water solution. It is worth mentioning that after HLD in all cases, there was not any discoloration of samples and liquid after each disinfection period. Hence, it could be stated that color pigment does not affect material properties after different types of HLD. The only one phenomenon of that kind of addition is the positive influence of chemical resistance.

The last type of sample was AM using the ABS medical. Strain–stress curves after tensile testing of that kind of material are shown in Figure 3.

ABS medical is characterized by a lack of significant influence of used HLD. The only one registered phenomenon is a slight decrease in the ultimate tensile strength (UTS) of samples held in 4% water solution, which could be connected with a hygroscopicity of the ABS. Similar insights were spotted in the DIC results shown in Table 7.

ABS medical material is characterized by a lack of any changes between samples before and after HLD in all used conditions. A fracture mechanism of all samples was similar–equal strain distribution in the UTS point and one, clearly visible fracture spot. Hence, that kind of material seems to be the most suitable for parts production which will be subjected to additional HLD to allow further usage from the tensile properties point of view.

Comparing all tested materials: PETG, colored PETG, and ABS medical, there are visible typical phenomena for those materials: ABS is characterized by 30% lower UTS than PETG with similar total strain values. In the case of comparison between colored PETG and noncolored there is visible a positive influence of pigment which make the material more resistant to disinfection but also lowered its UTS. Also, there are visible different phenomena registered during DIC where the PETG strain mechanism is strictly affected by the direction of material line distribution. At the same time in the ABS such behavior is local and the highest strain values are present in the necking area.

## 5. Summary and Conclusions

Based on obtained research results it could be stated that there is visible a slight influence of HLD on the material’s structure, its tensile properties, and strain mechanism. Regarding that kind of analysis, there were some changes registered that negatively affected PETG material. It has been proven that filament pigmentation does not affect material properties after HLD and does not cause any discoloration of the parts. Such phenomenon was observed during a longer analysis made by Buozi Moffa et al. [22] where there was also no discoloration registered. Using other types of disinfection for polymers such as photodynamic treatment [23] or microwaves [24] would negatively affect the mechanical properties of obtained parts by extending glass temperature or even plasticizing temperature which causes material degradation [25,26,27]. Such phenomena affect the material structure and also decrease its fatigue properties [28,29,30].

Regarding conducted research, the authors could form the following conclusions:AM technologies could be used to produce human life and health protection parts for sudden, unexpected cases. However, the layered structure of the obtained parts during the AM and some tiny pores between the infill and outline lines connection increases the possibility of penetration of that kind of imperfections by some fluids or bacteria strains. Form two types of tested materials, ABS medical seems to be a better candidate to produce such parts using AM.Pure PETG material is exposed to an even 20% decrease in tensile strength after HLD.Addition of color pigment in PETG material does not affect parts discoloration or decrease in tensile properties. It even slightly increases the material’s chemical resistance.Registered phenomenon with tensile strength decreasing observed in PETG samples could be related to the different alcohol diffusivity and solubility in these two materials. Additionally, the presence of the alcohol between extruded material lines could affect the joint volume between those lines. Another important issue is the fact that used in the research ABS material was dedicated for medical solutions—so its chemical resistance was increased to allow proper disinfection. PETG was a typical material available in the market which was not adopted for such a solution, but during the pandemic, it was the most popular material used in AM of tools dedicated for a fight against COVID19.

The next step in our research would be to investigate how long the liquid will stay inside the sample. If it would be a few hours, it would not impact drastically the usage daily. This is also an important question of whether the disinfectant reacts with the material. Further examination is required.

To reduce the presence of revealed voids in the structure of the parts a kind of heat treatment with an additional vacuum (like hot isostatic pressing in AM metals) could be used, which make it possible to fit that kind of technology for medical solutions and broader usage in that kind of application which could be another topic for further research.

## Figures and Tables

**Figure 1 materials-14-04823-f001:**
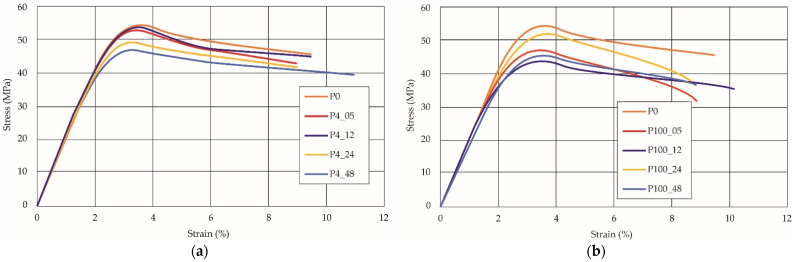
Strain-stress curves of noncolored PETG material after 4% water solution (**a**) and 100% concentrate (**b**) in comparison to material without disinfection.

**Figure 2 materials-14-04823-f002:**
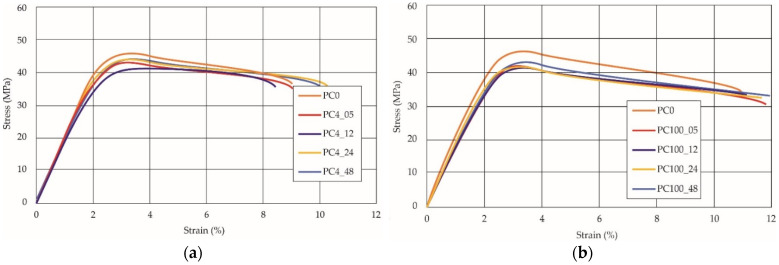
Strain–stress curves of colored PETG material after 4% water solution (**a**) and 100% concentrate (**b**) in comparison to material without disinfection.

**Figure 3 materials-14-04823-f003:**
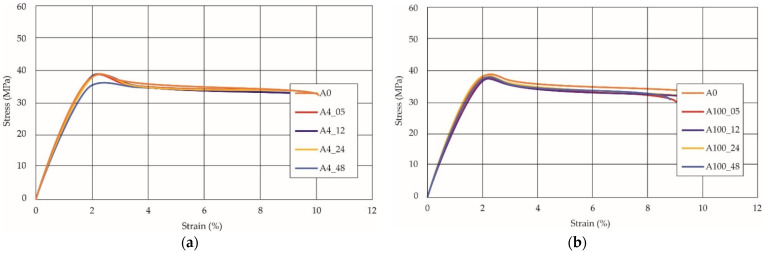
Strain–stress curves of colored PETG material after 4% water solution (**a**) and 100% concentrate (**b**) in comparison to material without any disinfection.

**Table 1 materials-14-04823-t001:** Properties of used materials: PETG and ABS medical provided by producers’ data sheets.

Material	PETG	ABS
Material density (g/cm^3^)	1.27	1.05
Flexural modulus (MPa)	2100	2600
Flexural strength (MPa)	69	75
Thermal deflection temperature (°C)	70	98
Vicat softening temperature (°C)	85	101

**Table 2 materials-14-04823-t002:** All configurations of samples prepared for the research.

Material	Solution Concentration	Disinfection Time	Samples’ Description
PETG without pigment	0%	None	P0
	4%	0.5 h	P4_05
		12 h	P4_12
		24 h	P4_24
		48 h	P4_48
	100%	0.5 h	P100_05
		12 h	P100_12
		24 h	P100_24
		48 h	P100_48
PETG with color pigment	0%	None	PC0
	4%	0.5 h	PC4_05
		12 h	PC4_12
		24 h	PC4_24
		48 h	PC4_48
	100%	0.5 h	PC100_05
		12 h	PC100_12
		24 h	PC100_24
		48 h	PC100_48
ABS Medical	0%	None	A0
	4%	0.5 h	A4_05
		12 h	A4_12
		24 h	A4_24
		48 h	A4_48
	100%	0.5 h	A100_05
		12 h	A100_12
		24 h	A100_24
		48 h	A100_48

**Table 3 materials-14-04823-t003:** Microscopic images registered for samples kept in 4% solution.

Microscope Images				
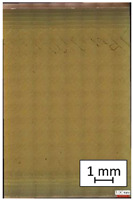 A0	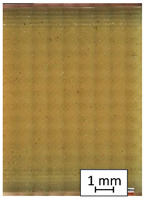 A4_0.5	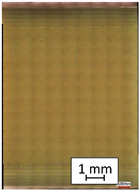 A4_12	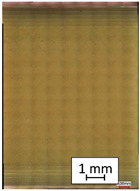 A4_24	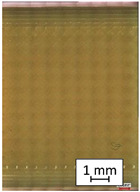 A4_48
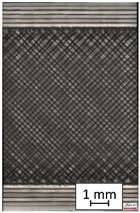 P0	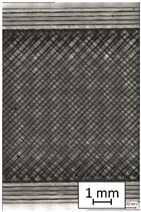 P4_0.5	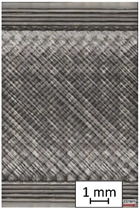 P4_12	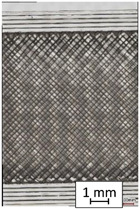 P4_24	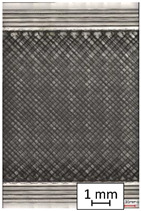 P4_48
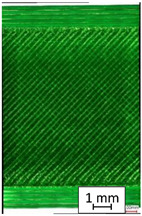 PC0	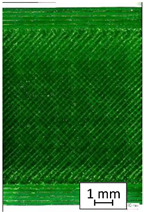 PC4_0.5	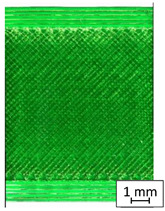 PC4_12	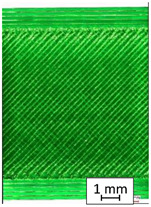 PC4_24	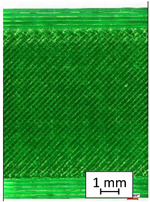 PC4_48

**Table 4 materials-14-04823-t004:** Microscopic images registered for samples kept in 100% solution.

Microscope Images				
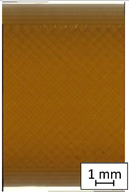 A0	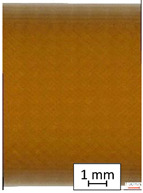 A100_0.5	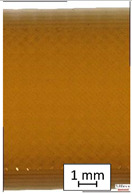 A100_12	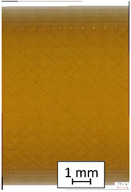 A100_24	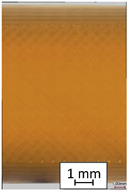 A100_48
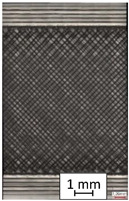 P0	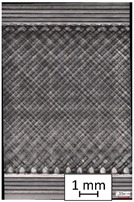 P100_0.5	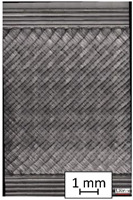 P100_12	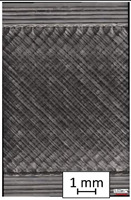 P100_24	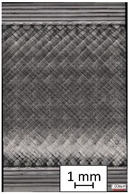 P100_48
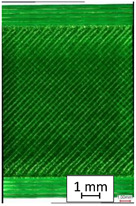 PC0	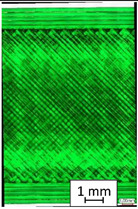 PC100_0.5	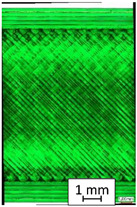 PC100_12	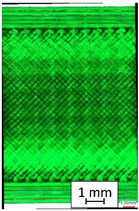 PC100_24	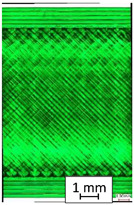 PC100_48

**Table 5 materials-14-04823-t005:** DIC results for without pigment PETG samples in the condition without HLD, in 4% water solution HLD and 100% concentrate HLD.

**P0**									
**Initial Condition**		**R_p0.2_**		**R_m_**		**Breaking Point**		**Fracture**	**Scale**
Strain X	Strain Y	Strain X	Strain Y	Strain X	Strain Y	Strain X	Strain Y		
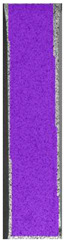	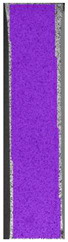	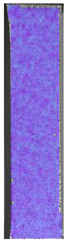	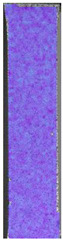	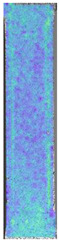	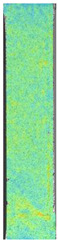	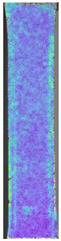	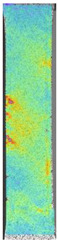	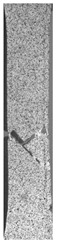	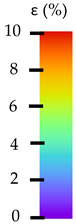
**P4_05**									
**Initial Condition**		**R_p0.2_**		**R_m_**		**Breaking Point**		**Fracture**	**Scale**
Strain X	Strain Y	Strain X	Strain Y	Strain X	Strain Y	Strain X	Strain Y		
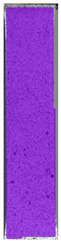	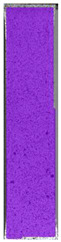	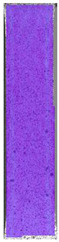	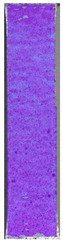	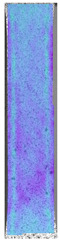	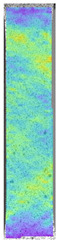	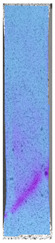	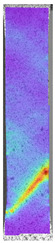	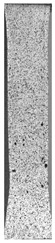	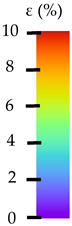
**P4_12**									
**Initial Condition**		**R_p0.2_**		**R_m_**		**Breaking Point**		**Fracture**	**Scale**
Strain X	Strain Y	Strain X	Strain Y	Strain X	Strain Y	Strain X	Strain Y		
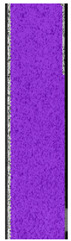	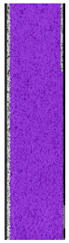	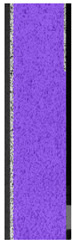	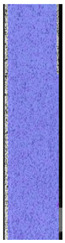	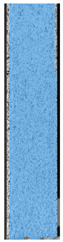	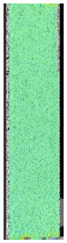	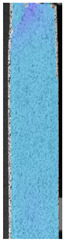	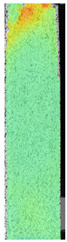	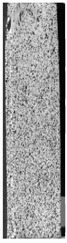	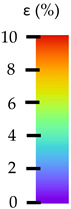
**P4_24**									
**Initial Condition**		**R_p0.2_**		**R_m_**		**Breaking Point**		**Fracture**	**Scale**
Strain X	Strain Y	Strain X	Strain Y	Strain X	Strain Y	Strain X	Strain Y		
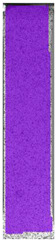	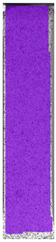	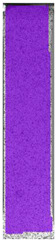	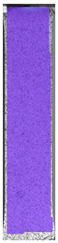	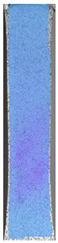	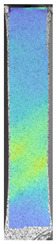	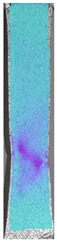	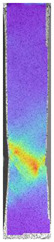	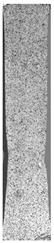	
**P4_48**									
**Initial Condition**		**R_p0.2_**		**R_m_**		**Breaking Point**		**Fracture**	**Scale**
Strain X	Strain Y	Strain X	Strain Y	Strain X	Strain Y	Strain X	Strain Y		
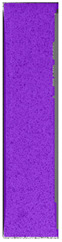	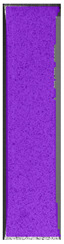	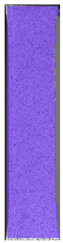	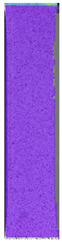	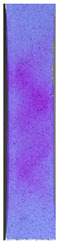	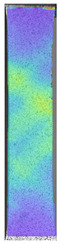	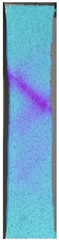	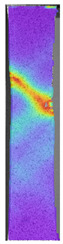	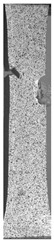	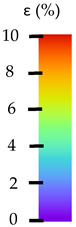
**P100_05**									
**Initial Condition**		**R_p0.2_**		**R_m_**		**Breaking Point**		**Fracture**	**Scale**
Strain X	Strain Y	Strain X	Strain Y	Strain X	Strain Y	Strain X	Strain Y		
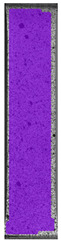	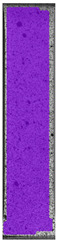	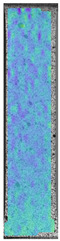	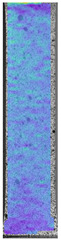	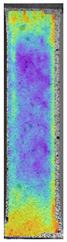	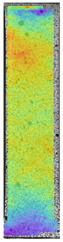	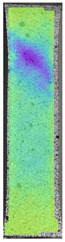	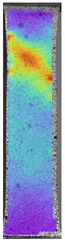	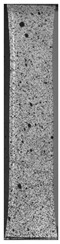	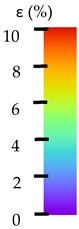
**P100_12**									
**Initial Condition**		**R_p0.2_**		**R_m_**		**Breaking Point**		**Fracture**	**Scale**
Strain X	Strain Y	Strain X	Strain Y	Strain X	Strain Y	Strain X	Strain Y		
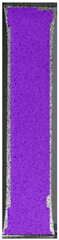	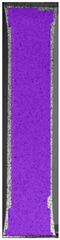	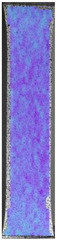	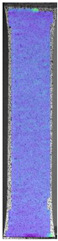	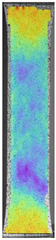	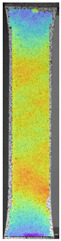	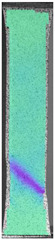	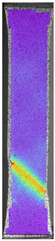	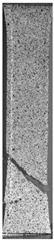	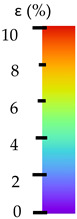
**P100_24**									
**Initial Condition**		**R_p0.2_**		**R_m_**		**Breaking Point**		**Fracture**	**Scale**
Strain X	Strain Y	Strain X	Strain Y	Strain X	Strain Y	Strain X	Strain Y		
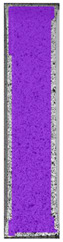	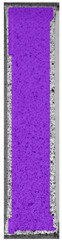	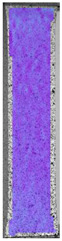	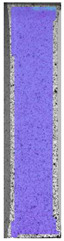	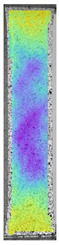	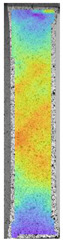	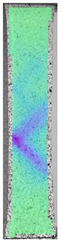	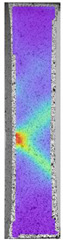	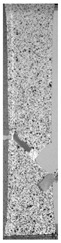	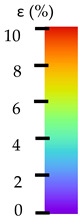
**P100_48**									
**Initial Condition**		**R_p0.2_**		**R_m_**		**Breaking Point**		**Fracture**	**Scale**
Strain X	Strain Y	Strain X	Strain Y	Strain X	Strain Y	Strain X	Strain Y		
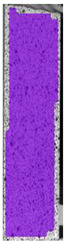	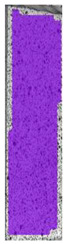	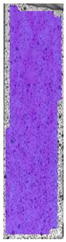	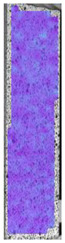	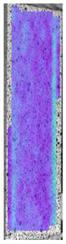	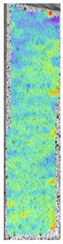	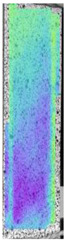	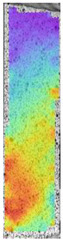	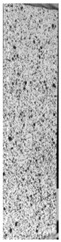	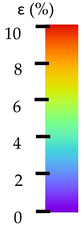

**Table 6 materials-14-04823-t006:** DIC results for colored PETG samples in the condition without HLD, in 4% water solution HLD and 100% concentrate HLD.

**PC0**									
**Initial Condition**		**R_p0.2_**		**R_m_**		**Breaking Point**		**Fracture**	**Scale**
Strain X	Strain Y	Strain X	Strain Y	Strain X	Strain Y	Strain X	Strain Y		
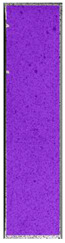	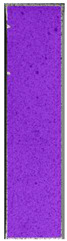	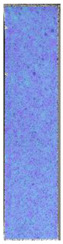	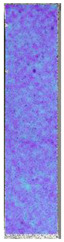	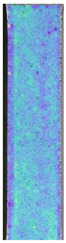	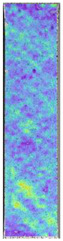	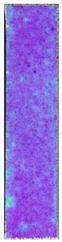	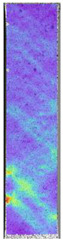	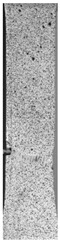	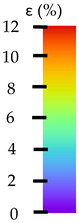
**PC4_05**									
**Initial Condition**		**R_p0.2_**		**R_m_**		**Breaking Point**		**Fracture**	**Scale**
Strain X	Strain Y	Strain X	Strain Y	Strain X	Strain Y	Strain X	Strain Y		
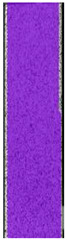	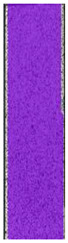	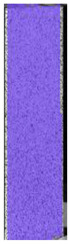	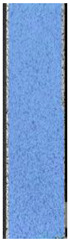	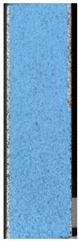	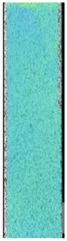	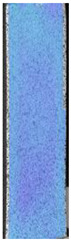	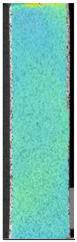	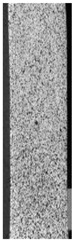	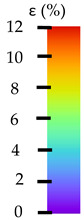
**PC4_12**									
**Initial Condition**		**R_p0.2_**		**R_m_**		**Breaking Point**		**Fracture**	**Scale**
Strain X	Strain Y	Strain X	Strain Y	Strain X	Strain Y	Strain X	Strain Y		
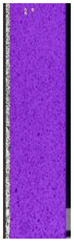	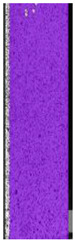	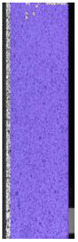	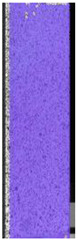	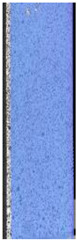	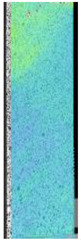	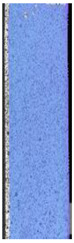	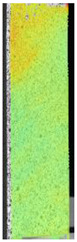	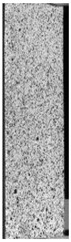	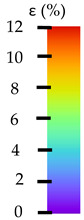
**PC4_24**									
**Initial Condition**		**R_p0.2_**		**R_m_**		**Breaking Point**		**Fracture**	**Scale**
Strain X	Strain Y	Strain X	Strain Y	Strain X	Strain Y	Strain X	Strain Y		
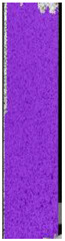	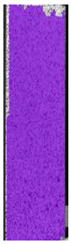	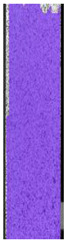	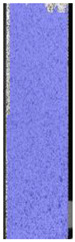	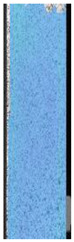	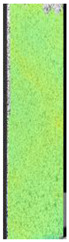	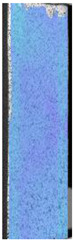	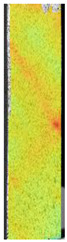	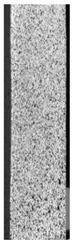	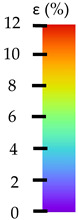
**PC100_05**									
**Initial Condition**		**R_p0.2_**		**R_m_**		**Breaking Point**		**Fracture**	**Scale**
Strain X	Strain Y	Strain X	Strain Y	Strain X	Strain Y	Strain X	Strain Y		
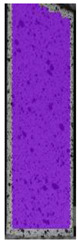	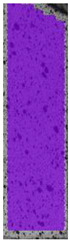	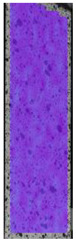	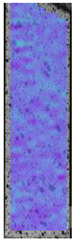	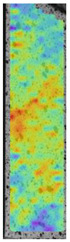	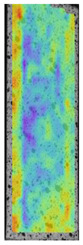	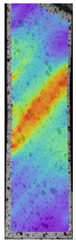	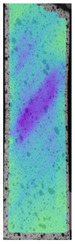	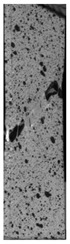	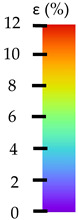
**PC100_12**									
**Initial Condition**		**R_p0.2_**		**R_m_**		**Breaking Point**		**Fracture**	**Scale**
Strain X	Strain Y	Strain X	Strain Y	Strain X	Strain Y	Strain X	Strain Y		
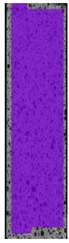	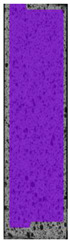	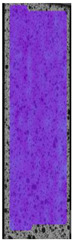	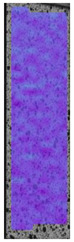	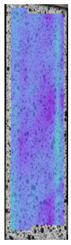	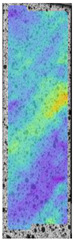	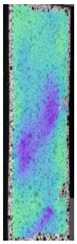	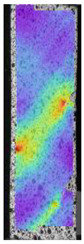	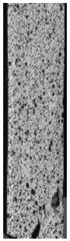	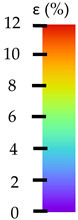
**PC100_24**									
**Initial Condition**		**R_p0.2_**		**R_m_**		**Breaking Point**		**Fracture**	**Scale**
Strain X	Strain Y	Strain X	Strain Y	Strain X	Strain Y	Strain X	Strain Y		
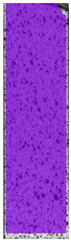	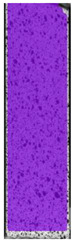	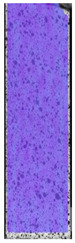	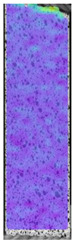	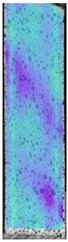	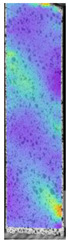	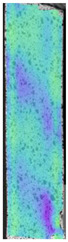	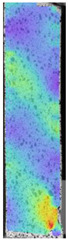	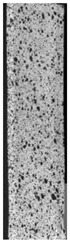	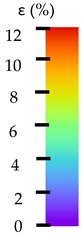
**PC100_48**									
**Initial Condition**		**R_p0.2_**		**R_m_**		**Breaking Point**		**Fracture**	**Scale**
Strain X	Strain Y	Strain X	Strain Y	Strain X	Strain Y	Strain X	Strain Y		
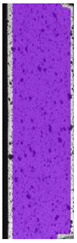	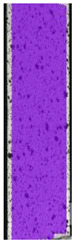	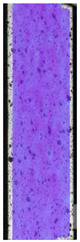	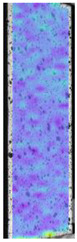	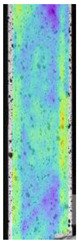	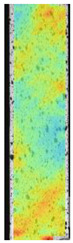	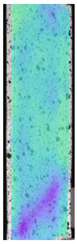	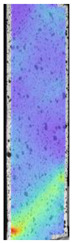	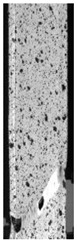	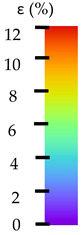

**Table 7 materials-14-04823-t007:** DIC results for ABS medical samples in the condition without HLD, in 4% water solution HLD and 100% concentrate HLD.

**A0**									
**Initial Condition**		**R_p0.2_**		**R_m_**		**Breaking Point**		**Fracture**	**Scale**
Strain X	Strain Y	Strain X	Strain Y	Strain X	Strain Y	Strain X	Strain Y		
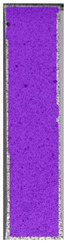	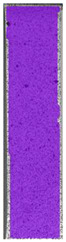	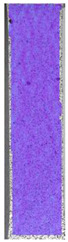	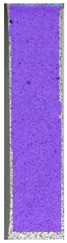	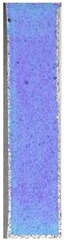	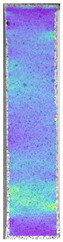	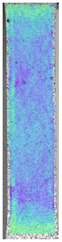	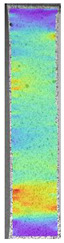	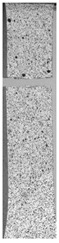	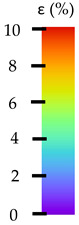
**A4_05**									
**Initial Condition**		**R_p0.2_**		**R_m_**		**Breaking Point**		**Fracture**	**Scale**
Strain X	Strain Y	Strain X	Strain Y	Strain X	Strain Y	Strain X	Strain Y		
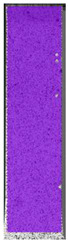	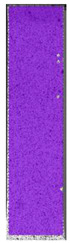	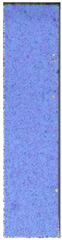	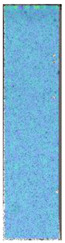	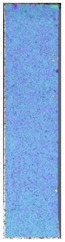	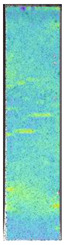	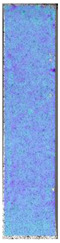	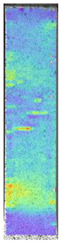	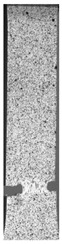	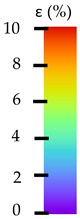
**A4_12**									
**Initial Condition**		**R_p0.2_**		**R_m_**		**Breaking Point**		**Fracture**	**Scale**
Strain X	Strain Y	Strain X	Strain Y	Strain X	Strain Y	Strain X	Strain Y		
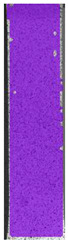	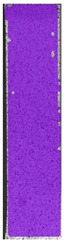	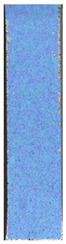	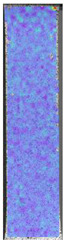	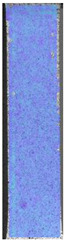	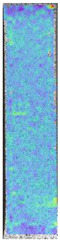	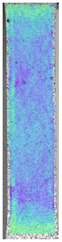	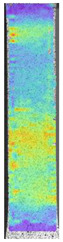	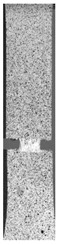	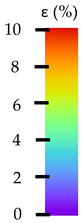
**A4_24**									
**Initial Condition**		**R_p0.2_**		**R_m_**		**Breaking Point**		**Fracture**	**Scale**
Strain X	Strain Y	Strain X	Strain Y	Strain X	Strain Y	Strain X	Strain Y		
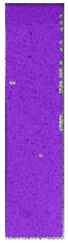	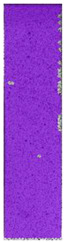	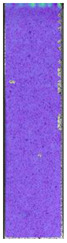	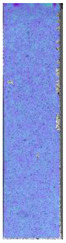	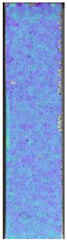	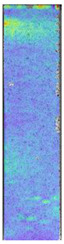	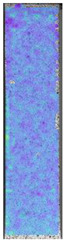	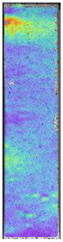	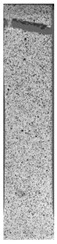	
**A4_48**									
**Initial Condition**		**R_p0.2_**		**R_m_**		**Breaking Point**		**Fracture**	**Scale**
Strain X	Strain Y	Strain X	Strain Y	Strain X	Strain Y	Strain X	Strain Y		
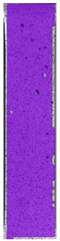	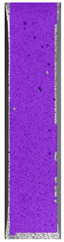	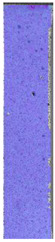	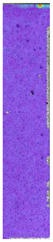	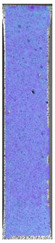	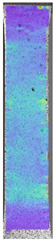	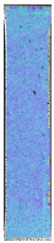	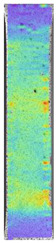	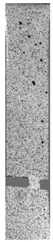	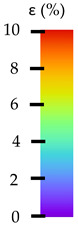
**A100_05**									
**Initial Condition**		**R_p0.2_**		**R_m_**		**Breaking Point**		**Fracture**	**Scale**
Strain X	Strain Y	Strain X	Strain Y	Strain X	Strain Y	Strain X	Strain Y		
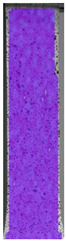	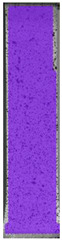	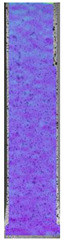	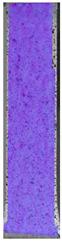	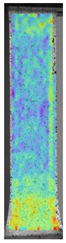	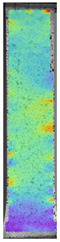	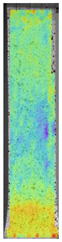	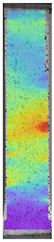	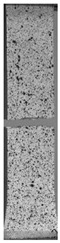	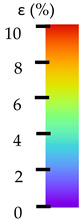
**A100_12**									
**Initial Condition**		**R_p0.2_**		**R_m_**		**Breaking Point**		**Fracture**	**Scale**
Strain X	Strain Y	Strain X	Strain Y	Strain X	Strain Y	Strain X	Strain Y		
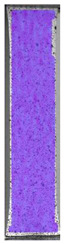	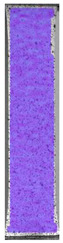	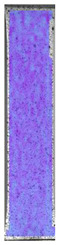	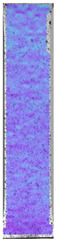	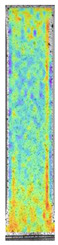	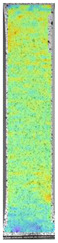	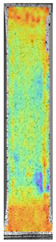	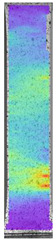	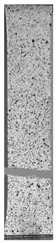	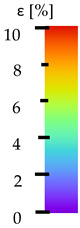
**A100_24**									
**Initial Condition**		**R_p0.2_**		**R_m_**		**Breaking Point**		**Fracture**	**Scale**
Strain X	Strain Y	Strain X	Strain Y	Strain X	Strain Y	Strain X	Strain Y		
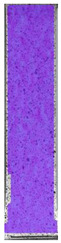	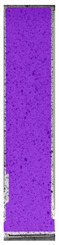	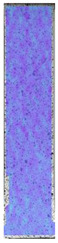	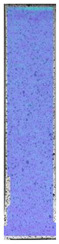	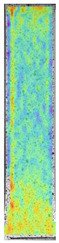	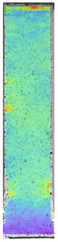	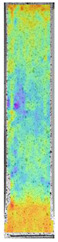	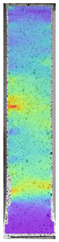	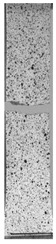	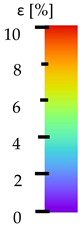
**A100_48**									
**Initial Condition**		**R_p0.2_**		**R_m_**		**Breaking Point**		**Fracture**	**Scale**
Strain X	Strain Y	Strain X	Strain Y	Strain X	Strain Y	Strain X	Strain Y		
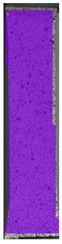	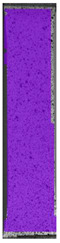	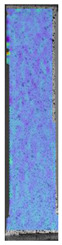	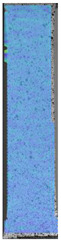	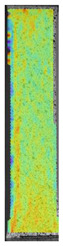	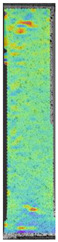	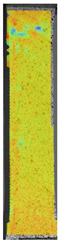	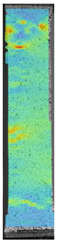	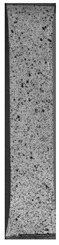	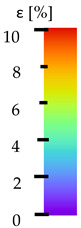

## Data Availability

The study did not report any data.

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
