# Peer review of "Additive Manufacturing of Plastics Used for Protection against COVID19—The Influence of Chemical Disinfection by Alcohol on the Properties of ABS and PETG Polymers"

_materials, 2021, doi:10.3390/ma14174823_

Round 1
Reviewer 1 Report
In this paper, the authors have investigated the influence of chemical disinfection on the structural properties of the components manufactured by the fused filament fabrication process. The results are interesting for the additive manufacturing community but before publication following comments should be addressed.
- The title of the paper seems to be misleading. The paper is more towards tensile strength analysis so it should be “chemical disinfection on structural properties.
- Line 16: HEPA used for the first time. Include its full form.
- Use the same abbreviations throughout the manuscript. For example, Polyethylene Terephthalate is initially defined as PETG but in the manuscript, the abbreviation is changed to PET-G.
- While defining abbreviations, the first letter of each should be capital. For example, instead of Material extrusion additive manufacturing (ME-AM), it should be Material Extrusion Additive Manufacturing (ME-AM). Please check the manuscript thoroughly for this.
- Line 105 – 112: How was the value of the parameters identified? Were they default for printers? Please state the reason behind selecting these values. The same comment applies from Line 120 – 127.
- Line 149: State the reason for selecting oven temperature as 45o
- In Table 1, for ABS medical 0.5h and 12h have the same sample description.
- In Table 2 and Table 3, It would be easy for the readers if you could give the sample description title same as in Table 1. For example, in table 1 title is P4_05 while in table 2 it is A_0.5_4%. It was difficult to relate the configuration of the sample with their microscopic images.
- Line 193 – 207: Need to be reframed. It looks like the authors are giving generalized information about images in Table 2 and Table 3. A lot of parts have been left for the reader's imagination. Whether the explanation for PETG is for with or without pigment? Tables need to be cited in the text as well.
- The authors say on line 199 that, “Earlier mentioned phenomena are connected to the lower dilution of used liquid” if so, how much was the dilution?
- Change the sample description in Tables 4, 5, and 6 according to Table 1.
- In conclusion, in Line 309, the authors mentioned materials structure. However, the paper is more about tensile properties and strain mechanisms, and no material structure was found.
- Point 4 of the conclusion is just the author's opinion and no hypothesis related to this conclusion was found in the manuscript. I think it should be included as future work.
Author Response
Dear Reviewer,
In the beginning, we would like to thank you for your valuable comments. We made proper corrections based on your comments. All changes made in our manuscript which were made according to your comments were yellow-highlighted. Regarding each paragraph:
- The title of the paper seems to be misleading. The paper is more towards tensile strength analysis so it should be “chemical disinfection on structural properties.
Answer: based on your advice we rephrased the title and now it has the following form: “Additive manufacturing of plastics used for protection against COVID19 - the influence of chemical disinfection on the properties of ABS and PETG polymers” - Line 16: HEPA used for the first time. Include its full form.
Answer: Thank you very much for your comment – we put a proper description.
- Use the same abbreviations throughout the manuscript. For example, Polyethylene Terephthalate is initially defined as PETG but in the manuscript, the abbreviation is changed to PET-G.
Answer: we unified it
- While defining abbreviations, the first letter of each should be capital. For example, instead of Material extrusion additive manufacturing (ME-AM), it should be Material Extrusion Additive Manufacturing (ME-AM). Please check the manuscript thoroughly for this.
Answer: Thank you very much for your advice – we made it
- Line 105 – 112: How was the value of the parameters identified? Were they default for printers? Please state the reason behind selecting these values. The same comment applies from Line 120 – 127.
Answer: Those parameters were a default setting for that kind of material – we put proper information about it and yellow-highlighted it.
- Line 149: State the reason for selecting oven temperature as 45o
Answer: Such temperature was selected to be at the safe level below the glass temperature of the PETG (which is 70°C). All samples were held at the same temperature. We put this information in the text.
- In Table 1, for ABS medical 0.5h and 12h have the same sample description
Answer: Thank you very much for this comment. We made proper corrections.
- In Table 2 and Table 3, It would be easy for the readers if you could give the sample description title the same as in Table 1. For example, in table 1 title is P4_05 while in table 2 it is A_0.5_4%. It was difficult to relate the configuration of the sample with their microscopic images.
Answer: You are 100% right – we unified it, thank you.
- Line 193 – 207: Need to be reframed. It looks like the authors are giving generalized information about images in Table 2 and Table 3. A lot of parts have been left for the reader's imagination. Whether the explanation for PETG is for with or without pigment? Tables need to be cited in the text as well.
Answer: We divided the description into each table to make it easier to interpret.
- The authors say on line 199 that, “Earlier mentioned phenomena are connected to the lower dilution of used liquid” if so, how much was the dilution?
Answer: We meant higher concentration (100%) now, after dividing it into two separate parts it is more understandable.
- Change the sample description in Tables 4, 5, and 6 according to Table 1.
Answer: It has been corrected.
- In conclusion, in Line 309, the authors mentioned materials structure. However, the paper is more about tensile properties and strain mechanisms, and no material structure was found.
- Answer: In the material structure phrase we meant the layered structure obtained during the AM process which is full of small pores – we rephrased it and name it as part/ sample structure obtained by AM
- Point 4 of the conclusion is just the author's opinion and no hypothesis related to this conclusion was found in the manuscript. I think it should be included as future work.
Answer: Thank you for that opinion. We put it in the part connected with further research.
We hope our corrections and explanations meet your expectations. We did our best to fit our manuscript for your and other reviewers’ comments. Thank you very much for your valuable and helpful comments.
With kind regards,
Authors
Reviewer 2 Report
The manuscript titled "Additive manufacturing of plastics used for protection against COVID19- the influence of chemical disinfection on structural and mechanical properties" (materials-1309042) describes the evaluation of the effect of chemical disinfection on the mechanical properties of additive manufactured materials usually used for COVID19 protection devices. The need of studies of this kind is well documented at the introduction where a brief explanation concerning additive manufacturing and fused filament fabrication is provided. Two different materials are compared, PETG and medical ABS, with two alcohol based solutions with varying concentration and exposure time.
However, the manuscript lacks characterization techniques. Authors correctly state that additive manufactured samples present are porous which influences the access and flux of solutions when materials are immersed. In addition, no degradation studies are presented even for the longer exposure times. It was observed by the authors that during the mechanical tests, the disinfection solutions were still entrapped inside the testing specimens so it would be of major importance to perform the mechanical tests after drying the disinfection solutions, and by doing that, to evaluate the real effect of the disinfection on the structural integrity of the specimens without the influence of the presence of the aqueous solutions.
Some chemical characterization is also lacking so it is quite difficult to state whether the disinfection solution jeopardizes the mechanical performance of the devices.
Lastly, there are no references in the discussion section,even with the increase available literature on mechanical properties of additive manufactured samples. Out of 21 references, 6 refer to autocitations where 5 are related to selective laser melting, which seems overstated, and only one corresponds to FFF related work.
Considering the overall submission, I do not recommend the publication of this manuscript in the journal "Materials". I advice the authors to develop the subject and then resumitt the manuscript as the topic is actual and of main interest.
Author Response
Dear Reviewer,
In the beginning, we would like to thank you for taking the time to read our manuscript. We agree that such research could be extended by many important data about analyzed materials (degradation studies, chemical investigation, etc.). We have focused mostly on tensile analysis and tried to make it as much detailed as it was possible. The first reason that we did not extend our research was a lack of proper laboratory devices. On the other hand - our manuscript has twenty pages right now, so doing the additional tests will make it too big for the readers.
Regarding your comment about drying - please have a look at line 149 (in uploaded corrected version), we put the following information: "After the process, all elements were drained into the laboratory oven at a temperature of 45°C for one hour. " Of course such information was also in the original version.
We wish we could have proper laboratory devices to attach results after chemical analysis. Regarding the fact that we are a team from the mechanical engineering department all that we were able to do was detailed tensile tests extended by digital image correlation and microscopical investigation, so we did our best to describe the material. To prepare such analysis with three materials, two concentrations, and five-time regimes took a lot of time and people work to make it as good as possible.
Based on your comment we extend the discussion and put more citations about the materials and technology.
This topic would be continued in further research. But this manuscript after extension would take too many pages. On the other hand, removing some parts to put additional results could make it worse.
We hope you will understand our point and find our paper valuable enough to publish it in the Materials.
Yours sincerely,
Authors
Reviewer 3 Report
Grzelak et al. studied the influence of alcohol disinfection on the structural, mechanical properties of FDM printed ABS and PETG polymers. Three types of materials were tested in four different disinfection times and two disinfection alcohol concentrations. It was found the disinfection liquid penetrated into the material’s voids and led to some reduction of mechanical properties of the printed PETG parts. This timely work is helpful in leveraging 3D printed polymer parts to fight against the COVID19 pandemic. I recommend accepting the manuscript after addressing the following questions.
- Some content and discussion for the mechanistic understanding of alcohol disinfection on the mechanical properties can be added. For example, larger magnification microscope images can be provided to show the morphology change at the filament level.
- Both the modulus and strength of the treated PETG show a minor decrease. The reduction in mechanical properties can be attributed to either the material degradation or plasticization by the alcohol. Relevant experiments can be added to identify the major reason. For example, Fourier-transform infrared spectroscopy can be used to detect if there is residual alcohol in the treated material. Or samples with a longer drying time after treatment can be used for comparison.
- More discussion on the different influences of disinfection on PETG and ABS can be added. The possible reason can be the different alcohol diffusivity and solubility in these two materials. A swelling test experiment or some, and references can be added.
- Page 3, line 130, correct the front quotation mark.
Author Response
Dear Reviewer,
In the beginning, we would like to thank you for your valuable comments. We made proper corrections based on your comments. All changes made in our manuscript which were made according to your comments were yellow-highlighted. Regarding each paragraph:
- “Some content and discussion for the mechanistic understanding of alcohol disinfection on the mechanical properties can be added. For example, larger magnification microscope images can be provided to show the morphology change at the filament level.”
It is a very important phenomenon that we tried to analyze. Unfortunately, a larger magnification did not expose any additional changes in the material structure. We also tried to grind and make more advanced material investigation with SEM analysis. During grinding the disinfection liquid gets out from the pores in the ground surface. Also, there were not visible any changes in the material structure – that is why we did not attach results from this research. Following the advice from the other reviewer, we would like to do make further research in cooperation with chemical researchers who will help us to analyze changes based on polymer’s chemical composition.
- “Both the modulus and strength of the treated PETG show a minor decrease. The reduction in mechanical properties can be attributed to either the material degradation or plasticization by the alcohol. Relevant experiments can be added to identify the major reason. For example, Fourier-transform infrared spectroscopy can be used to detect if there is residual alcohol in the treated material. Or samples with a longer drying time after treatment can be used for comparison.”
Thank you for this comment. We have focused mostly on tensile analysis and tried to make it as much detailed as it was possible. The main reason that we did not extend our research was a lack of proper laboratory devices. Regarding longer drying time we use only one configuration because we created a lot of combinations with disinfection time and liquid concentration. Such combinations exposed issues that should be subjected to further analysis. Drying time has been taken from the typically used configuration of 3D printed parts for a fight against COVID19. You mentioned very important issues which we want to do in our further research to describe the degradation mechanism of AM polymers. At this level, we tried to do our best to describe material behavior during tensile testing including stain analysis. We hope it is enough for this publication.
- More discussion on the different influences of disinfection on PETG and ABS can be added. The possible reason can be the different alcohol diffusivity and solubility in these two materials. A swelling test experiment or some, and references can be added.
We put the additional discussion in the 4th point of summary: “Registered phenomenon with tensile strength decreasing observed in PETG samples could be related to the different alcohol diffusivity and solubility in these two materials. Additionally, the presence of the alcohol between extruded material lines could affect the joint volume between those lines. Another important issue is the fact that used in the research ABS material was dedicated for medical solutions – so its chemical resistance was increased to allow proper disinfection. PETG was a typical material available in the market which was not adopted for such a solution, but during the pandemic, it was the most popular material used in AM of tools dedicated for a fight against COVID19”.
- “Page 3, line 130, correct the front quotation mark.”
It has been corrected, thank you.
We hope our corrections and explanations meet your expectations. We did our best to fit our manuscript for your and other reviewers’ comments. Thank you very much for your valuable and helpful comments.
With kind regards,
Authors
Reviewer 4 Report
This manuscript present a very important problem of materials that is placed on the top of significance. The authors found an interesting way to characetize the studied materials. However, a lot of information is absent: the behaviour on absorbing liquids (swelling), the durability of items, the penetration rate, the comparative analysis of presented results.
Some important aspects must be improved:
- the Abstract is not a place where the general point of viwe is presntes; it must provide the most important rsults,
- why PLA is detailed analyzed in Introduction, but not studied materials,
- several characteristics that recommend them for care-help must be discused togheter with the functional properties,
- the stress-strength curves are not comparatively discussed on different materials but the same concentration of solution,
I do not understand why the authors nominate photo-presentation as tables that, in fact, they are photos; why some parts are highlighted.
In spite of an excellent English, the academic feature is difficult to be detected. I suggest a profound improvement of presentation, including several defining material characteristics.
The authors must follow the instruction for authors (template) of this journal.
Author Response
Dear Reviewer,
In the beginning, we would like to thank you for your valuable comments. We made proper corrections based on your comments. All changes made in our manuscript which were made according to your comments were yellow-highlighted. Regarding each paragraph:
- “the Abstract is not a place where the general point of view is presntes; it must provide the most important rsults,”
Thank you for this comment. We removed the part with a general point of view and rewritten the abstract.
- “why PLA is detailed analyzed in Introduction, but not studied materials,”
We detailed the PLA because it is one of the most important materials in the ME-AM. To justify why we did not analyze it we put an additional sentence after its description and cover it using proper citation: “That is why this material has not been used during AM of parts dedicated for protection against COVID19 [2]”.
- “several characteristics that recommend them for care-help must be discused togheter with the functional properties”
We put an additional description of two used materials in our research. It has been green highlighted in the text:
“The main reason for using PETG for care-help during the beginning of the COVID19 pandemic is twofold. The first is related to the material properties – it is more resistant to temperature than the PLA – it can withstand temperatures up to 75°C and its properties are not affected by the UV radiation. Also, it is characterized by better chemical resistance than ABS and PLA and is less fragile than the PLA parts. Such characteristics allow bettering disinfection using high temperature or alcohols which made it better customized to medical solutions than the PLA and other commonly used materials in ME-AM. On the other hand, the usage of the PETG at the beginning of the COVID19 pandemic was related to low filament diameter tolerance increase after winding increase (prusament.com) to al-low the increase of manufactured material dedicated to FFF technology.”
“Additionally, there are available in the market, materials dedicated for the ME-AM, and usage in medical applications. The most popular are ABS-based filaments. Such materials are mostly certified by basing on USP VI and ISO 10993-1 standards. Its usage”
- “the stress-strength curves are not comparatively discussed on different materials but the same concentration of solution,”
We used such a strategy because we wanted to focus on the influence of disinfection on the properties of the material – not on the comparison of tensile strength of all tested materials. Such analyzes are widely described in the literature. We wanted to highlight how disinfection affects each material behaviour during uniaxial static loading. To make the description more comparatively discussed we put an additional paragraph at the end of chapter 4 and green highlighted it:
“Comparing all tested materials: PETG, colored PETG, and ABS medical, there are visible typical phenomena for those materials: ABS is characterized by 30% lower UTS than PETG with similar total strain values. In the case of comparison between coloured PETG and noncoloured there is visible a positive influence of pigment which make the material more resistant to disinfection but also lowered its UTS. Also, there are visible different phenomena registered during DIC where the PETG strain mechanism is strictly affected by the direction of material line distribution. At the same time in the ABS such behavior is local and the highest strain values are present in the necking area.”
- “I do not understand why the authors nominate photo-presentation as tables that, in fact, they are photos; why some parts are highlighted.”
We named it tables because we must divide it into proper cells where we were able to put material in exact strain conditions. Such distribution is more common for tables. Of course, we are ready to change it if you find our idea of dividing in the table wrong. We highlighted some parts for another reviewer to allow him to recognize corrected parts. It is similar to the correction which was green highlighted based on your comments.
- “In spite of an excellent English, the academic feature is difficult to be detected. I suggest a profound improvement of presentation, including several defining material characteristics.”
We carefully read the whole manuscript and tried to make it more academic. All used words for the description of samples and material characteristics are commonly used in literature and standards. We have checked it once again to be sure about it.
- “The authors must follow the instruction for authors (template) of this journal.”
We used the newest version of the template for the Materials during creating our manuscript.
We hope our corrections and explanations meet your expectations. We did our best to fit our manuscript for your and other reviewers’ comments. Thank you very much for your valuable and helpful comments.
With kind regards,
Authors
Round 2
Reviewer 2 Report
After considering the response from the authors, I still think that the paper would strongly benefit from revising.
I understand the limitations stated by the authors, however, I do not think that the experiment design is complete. In my opinion, when evaluating the effect of disinfection, a degradation test is almost mandatory. I understand that the document already is 20 pages long, but it could be overcome by placing some of the images as supporting information. Furthermore, the authors state that they wanted to focus on the tensile strength of the specimens, but, all tests were performed using the same conditions, which seems poor concerning that the topic is suppose to be the main subject of the paper.
I still believe that is a little bit overstated to auto cite 4 works concerning SLM of stainless steel 316L, considering that the present paper refers to FFF and polymeric materials, which should be the focus of the introduction section.
Considering the overall submission, I do not recommend the publication of this manuscript in the journal "Materials". I advise the authors to develop the subject and then resubmit the manuscript as the topic is actual and of main interest.
Author Response
Dear Reviewer,
As we mentioned before, if we will be able to make such an analysis as degradations tests we will add it. We tried to find a proper unit in our University but it was not possible to do it during the time for corrections after review. Thanks to your comments (and also other reviewers) we have prepared a list of research which we will use in further research included in the next manuscript.
Using the same conditions was necessary to make results comparable and check how disinfection affects material properties of each material (PETG, coloured PETG, ABS Medical). Additionally, there are no standards for the disinfection of AM materials. All disinfection conditions were taken from medical facilities (we asked what conditions they were used). Regarding tensile testing, we made all activities regarding the ASTM D638 standard which is commonly used in such experiments.
316L stainless steel is commonly used in many medical solutions. What is more, it is a recommended material for manufacturing tools to use in medicine. We used citations regarding the SLM technology to cover a statement about using cheaper technologies than AM of metals.
We find your comments very valuable but we still believe that our research is worth publication, because it presents very actual and important issues of using AM technologies in medical solutions and its pros and cons.
Independently of your decision, we would like to thank you for your valuable comments which helped us to improve our manuscript.
Yours sincerely,
Authors
Reviewer 3 Report
The authors didn't address the major comments properly. The reviewer understands some additional experimental may be difficult at the pandemic time. Some necessary experimental, at least some more in-depth discussion on the mechanism, including the morphology and chemical structures, can be added to improve the manuscript.
Additional comments:
- In the introduction, state of the art advances in chemical cleaning, chemical disinfection of FDM printed parts are lacking. Most of the content deal with the influence of printing parameters on the mechanical properties. The gap of previous works in chemical disinfection can be further highlighted.
- This work only deals with the chemical disinfection by alcohol on the mechanical properties of ABS and PETG. There are also many other chemical disinfection methods, including chlorines, iodine, Formaldehyde, and alcohol. Other general physical properties, including thermomechanical properties and heat resistance, etc. are not discussed. In this sense, the title should be adjusted to be more specific.
Author Response
Dear Reviewer,
Regarding your comments, we have made additional descriptions related to chemical disinfection mechanisms and the influence of different types of disinfection on additive manufactured parts. Additionally, we attached a table with both material properties (PETG and ABS) provided by materials producers' - table 1. All mentioned corrections we highlighted using blue colour. We hope you will find our improvements valuable and sufficient.
Reviewer 4 Report
The present version of manuscript in closer that the previous one on the academic paper. However, the Englisg style is still poor; it must be really improves. Some words are written with great letters, like PolyLactic Acid and Acrylonitrile Butadiene Styrene page 1, line 38), Universities (page 2 , line 87) a. s. o. The manuscript must be elaborate more carefully ( see "for care-help").
Author Response
Dear Reviewer,
We are a little bit confused because all faults which you mentioned were suggested by other reviewers.
1. " Some words are written with great letters, like PolyLactic Acid and Acrylonitrile Butadiene Styrene page 1, line 38)" - it was suggested by Reviewer 1: "While defining abbreviations, the first letter of each should be capital. For example, instead of Material extrusion additive manufacturing (ME-AM), it should be Material Extrusion Additive Manufacturing (ME-AM). Please check the manuscript thoroughly for this."
2. Universities (page 2, line 87) a. s. o. The manuscript must be elaborate more carefully ( see "for care-help") was used regarding your comment: "several characteristics that recommend them for care-help must be discussed together with the functional properties”. If in your opinion it should disappear - we replaced it already.